# Changes of Gene Expression Patterns from Aquatic Organisms Exposed to Metal Nanoparticles

**DOI:** 10.3390/ijerph18168361

**Published:** 2021-08-07

**Authors:** Mateusz Kulasza, Lidia Skuza

**Affiliations:** 1Institute of Biology, University of Szczecin, 71-415 Szczecin, Poland; lidia.skuza@usz.edu.pl; 2The Centre for Molecular Biology and Biotechnology, University of Szczecin, 71-415 Szczecin, Poland

**Keywords:** metal nanoparticles, gene expression, aquatic toxicology

## Abstract

Metal nanoparticles are used in various branches of industry due to their physicochemical properties. However, with intensive use, most of the waste and by-products from industries and household items, and from weathering of products containing nanoparticles, end up in the waters. These pollutants pose a risk to aquatic organisms, one of which is a change in the expression of various genes. Most of the data that focus on metal nanoparticles and their effects on aquatic organisms are about copper and silver nanoparticles, which is due to their popularity in general industry, but information about other nanoparticulate metals can also be found. This review aims to evaluate gene expression patterns in aquatic organisms by metal nanoparticles, specifying details about the transcription changes of singular genes and, if possible, comparing the changes in the expression of the same genes in different organisms. To achieve this goal, available publications tackling this problem are studied and summarized. Nanometals were found to have a modulatory effect on gene expression in different aquatic organisms. Data show both up-regulation and down-regulation of genes. Nano silver, nano copper, and nano zinc show a regulatory effect on genes involved in inflammation and apoptosis, cell cycle regulation and ROS defense as well as in general stress response and have a negative effect on the expression of genes involved in development. Nano gold, nano titanium, nano zinc, and nano iron tend to elevate the transcripts of genes involved in response to ROS, but also pro-apoptotic genes and down-regulate DNA repair-involved genes and anti-apoptotic-involved genes. Nano selenium showed a rare effect that is protective against harmful effects of other nanoparticles, but also induced up-regulation of stress response genes. This review focuses only on the effects of metal nanoparticles on the expression of various genes of aquatic organisms from different taxonomic groups.

## 1. Introduction

The growing use of nanomaterials pose a threat to the aquatic environment as factory efflux and waste from products in common use such as textiles enriched with nano silver or paint-containing nanometals, enter the water directly [1]. Nanoparticles (NPs) are particles that have a dimension from 1 to 100 nm with properties varying from those of bulk material of the same chemical composition [2,3]. Metal NPs are known to have toxic effects on aquatic organisms, which include changes in gene expression. 

There are many different materials that may be turned into nanoparticles. The classification based on the source material consists of carbon-based nanoparticles, such as fullerenes and nanotubes (NTs), ceramics NPs, metal NPs, and organic NPs, such as lipids, and all differ in size, shape, charge, surface modifications, etc. [4]. To obtain unnatural atomic structures, it is necessary to provide a special environment for the synthesis reaction, i.e., increasing the temperature or pressure [5].

The toxic effect of NPs on living organisms is induced by reactive oxygen species (ROS), the generation of which can be stimulated by various mechanisms of action of NPs [6,7]. The presence of prooxidative functional groups on the metal nanoparticles surface, active redox cycle of transition metals NPs and particle–cell interaction are the main factors in the induction of generation of ROS [7,8]. Apart from the surface properties, metal NPs can act as ROS production catalyzers by various mechanisms. Metal NPs are treated as a threat by immunocompetent cells that generate ROS as a defensive mechanism [6,7]. Transition metals can induce ROS generation by taking part in Haber–Weiss and Fenton-like reactions, some other NPs activate cellular pathways involved in ROS generation as signal transductors [8]. In mitochondria, metal NPs disrupt the electron transport chain, which leads to “electron leakage”; free electrons are then able to form ROS from the oxygen particles [9]. Another major action of nanoparticles on living organisms is the release of their components as free ions, which can have toxic properties. Additionally, copper released from copper nanoparticles can directly interact with DNA and disrupt its structure [10].

Most popular in various branches of industry are copper and silver nanoparticles due to their versatility and a wide range of interesting properties. For that reason, most of the related molecular studies tend to focus on the effects of those two types of metal nanoparticles [10,11].

The aim of this review is to study and compare changes in the gene expression of various genes from various aquatic organisms, which was achieved by a review of the available literature. 

## 2. Effects of Metal Nanoparticles on Gene Expression

### 2.1. Effects of Silver Nanoparticles on Gene Expression 

Silver nanoparticles (AgNPs) found a place in general industry (textile industry, cosmetics, pharmaceutics, paints etc.) mostly due to their antibacterial properties [12]. They are released to the environment mostly as efflux from factories, aerosols, and as waste from use and weathering products containing added AgNPs [4]. In the presence of organic matter in the water, AgNPs appear to be very stable [13].

Silver nanoparticles are known to be toxic to some aquatic plants and crustaceans [14,15]. It was found that AgNPs disrupt the proper development of *Oryzias latipes* larvae and *Danio rerio* embryogenesis [16,17,18,19]. 

In sea urchin, *Strongylocentrotus droebachiensis,* and in adult *Danio rerio,* nano silver triggers an immune response [20,21]. Apart from that, there are data stating that nano silver influences gene expression of aquatic organisms.

A study on the zebrafish embryos treated with silver nanocolloids (AgNCs) and silver nanotubes (AgNTs) caused both up-regulation and down-regulation of various genes [22]. AgNCs caused up-regulation of 98 genes and down-regulation of 166 genes from a total of 264 expressed genes, while embryos treated with AgNTs showed 36 up-regulated and 139 down-regulated genes from a total of 175 expressed genes. In the AgNCs group, the top 10 up-regulated genes were *zgc:**175127* (cellular component), *sort1b* (involved in lipid metabolism), *idh3g* (carbohydrate metabolism), *thrab* (developmental processes), *lgals3l* (lectin), *caspb* (proteolysis), *zgc:114104* (metal ion binding gene), *pvalb5* (calcium/calmodulin-binding and *lnx1* (notch signaling gene). Overexpression of some of these genes, such as *caspb* and *lgals3l,* is involved in inflammation and apoptosis. Genes down-regulated by AgNCs in zebrafish embryos included *per1b* (response to light stimulus), *nt5c3* (pyrimidine nucleoside metabolism), *mif4gdb* (regulation of translation), *cbx7a* (chromatin modification), *mep1a.2* (proteolysis), *slc45a3* (transmembrane transport), *myf5* (muscle organ development) and *irg1l* (propionate metabolism). 

The AgNTs-treated group mostly showed up-regulation of different and fewer genes, the top ten being *pepd* (peptidase), *sema3fb* (semaphoring 3fb), *crestin* (crestin), *zgc:103670*, *map2k2a* (mitogen-activated protein kinase 2a), *zgc:175127*, *anxa1b* (annexin A), *mylpfb* (myosin light chain), *fgf11a* (fibroblast growth factor 11a), and *cdh15* (cadherin 15), from which three are involved in calcium binding (*anxa1b*, *mylpfb*, *cdh15*). The down-regulated gene from the AgNTs group involved those that take part in DNA-dependent transcriptional regulation, cell proliferation, and differentiation, i.e., *tfdp2*, *fosb*, *nr1d1*, *jdp2*, *fox1a*, *nfil13-6,* and *fos*) [22]. In *Onorhynchus mykiss* exposed to silver nanoparticles Ag N_10_, one gene was found to be overexpressed, namely *cyp1a2,* involved in general metabolism [23].

*D.* rerio embryos exposed to 4nm AgNPs showed significantly up-regulated *hypoxia-inducible factor* 4 (HIF4; adaptation to the state of hypoxia) genes, which implies that after exposure, a state of hypoxia occurred in the embryos. Another up-regulated gene was that encoding peroxisomal membrane protein 2 (*Pxmp2*), whose product participates in a continual flow of small solutes [16]. Another form of AgNPs, nano-silver coating material, causes the *D. rerio* embryos to up-regulate the transcription of the transforming factor β (TGF-β) involved in cellular pathways, angiogenesis, apoptosis, and cell cycle, as well as signaling superfamily members *pitx2*, *acvr2b*, and *smad1,* crucial in forming various body structures; however, inhibin β B (*inhβb*), involved in oocyte differentiation, growth, and hormone activity, was down-regulated [19]. Another study performed on adult zebrafish liver after treatment with silver nanoparticles [24] also showed a decrease in the expression of oxyradical scavenging enzymes catalase, glutathione peroxidase 1a and superoxide dismutase 1, while induction of some proapoptotic genes occurred, i.e., *Bax1*, *Noxa*, and*p21*. 

Sublethal doses of AgNPs of 10 nm and 35 nm increased levels of MT2 (metallothioneins, involved in metal ions binding) in *D. rerio* [25]. Furthermore, both adult oysters and embryos had highly increased MT mRNA levels after being treated with AgNPs [26].

In zebrafish embryos 24 h post-fertilization treated with AgNPs, there was significant down-regulation of cytochrome 1a gene that is involved in circadian rhythm. Other differently transcribed genes included the down-regulated after 24 hpf (hours post fertilization) and up-regulated after 48 hpf proteasome subunit β type-1, which takes part in protein catabolism, and up-regulated ribosomal protein S6 modification-like protein B [18].

*Oncorhynchus mykiss* treated with silver nanoparticles at 0.1 mg L^−1^ and 0.25 mg L^−1^ revealed increased transcription levels of MT, GST (glutathione transferase), and SOD (superoxide dismutase), which are involved in the antioxidative defense system, in the gills compared to a control group in a dose-dependent manner [27]. While silver nanoproducts induced up-regulation of genes encoding proteins that take part in the antioxidant defense pathways, in adult zebrafish liver treated with AgNPs, some of the genes involved in antioxidant defense were down-regulated, whose protein products were MTF-1 (metal-responsive transcription factor), TLR4 (Toll-like receptor, pattern recognition receptor), IL1B (interleukin, immune response), CEPB (CCAAT/enhancer-binding protein beta, transcription factor), TRF (telomere-binding protein, formation of a T-loop), and TLR22 [21]. 

In the fourth-instar larvae of *Chironomus riparius,* exposure to 1mg L^−1^ AgNPs disrupts expression of the ribosomal protein gene L15 (*CrL15*), affecting the ribosomal assembly and at the same time induces up-regulation of the gonadotrophin-releasing hormone gene (*CrGnRH1*), which may lead to the over-activation of the gonadotrophin-mediated pathways and to reproductive failure, and Balbiani repeat gene 2.2 (*CrBR2.2*). Another study on *C. riparius* reported differently expressed *CrGST* genes after treatment with AgNPs for various times [28,29]. In the eggs, larvae, and pupae of *C. riparius,* AgNPs induced overexpression of Hsp70, a stress marker, but 0.2 mg L^−1^ after 24 and 48 h of exposure lowered the levels of *CrRPL15* (ribosomal protein L15, active role in the constitution of the active 50S subunit of the ribosome) transcript levels, like in the previously mentioned study [30]. 

After 28 days of exposure to 1 µg L^−1^ and 25 µg L^−1^ silver nanoparticles, the liver of male medaka (*Oryzias latipes*) showed increased expression of S-glutathione transferase gene for 1 µg L^−1^ concentration and Hsp70, choriogenin, and vitellogenin genes for 25 µg L^−1^ concentration, whereas both concentrations of silver NPs induced higher expression of transferrin than in the control group [31]. After mussels *Mytilus galloprovincialis* had been incubated for two weeks in water containing 10 µg L^−1^ AgNPs, it was found that CYP4YI (cytochrome) and cathepsin (protease) genes showed increased transcript levels, whereas GST and caspase genes (protease involved in apoptosis) exhibited down-regulation [32]. 

Similarly, in *S. droebachiensis* immune cells, increased transcription of Hsp70 and Hsp60 was observed after keeping them for 48 h in an environment containing 100 µg L^−1^ AgNPs [20].

Kwok et al. [17] tested different coatings of AgNPs and their effects on gene expression of orange-red medaka (*O. latipes*). Coatings included polyvinylpyrrolidone (PVP), gum arabic (GA), and citrate (Cit), either solely citrate, or conjugated with either tannic acid or glutathione.

All types of NPs and coatings caused up-regulation of CYP1a (stress response). Oxidative stress genes were up-regulated only by PVP coating and GA coating. NKA transcripts, however, were elevated by all types of AgNPs but those coated with Cit or Cit-Tannic acid. 

Wang et al. [33] tested the AgNPs on the expression of copper deficiency response genes in green algae *Chlamydomonas reinhardtii*, which turned out to be up-regulated during the treatment. These genes included CYC6 (heme-containing c-type cytochrome C6, involved in copper level regulation), FDX5 (ferredoxin, reductive metabolism in the chloroplast), CTR2 (copper assimilation, copper transporter), and *CRD1* (copper response defect 1). In the study by Simon et al. [34] on *C. reinhardtii*, AgNPs induced up-regulation of 141 genes and down-regulation of 86 genes. 

### 2.2. Effects of Copper Nanoparticles on Gene Expression

Copper nanoparticles (CuNPs) can be found in paints, optical and medical instruments, water pipes, conductive materials, and antibacterial coatings. Their release to the environment is caused, among other things, by factories releasing them as waste efflux, by aerosols, and by exploitation of the products and their weathering [1].

Nano copper is a known toxicant towards aquatic organisms. It was noted to cause histopathological changes in rainbow trout (*O. mykiss*) [35] and gill injury and acute lethality in *D. rerio* [11], and injuries to liver and gills were found in *Epinephelus coioides* after CuNPs treatment [36] and in the gills of *Solea senegalensis* [37]. Toxic effects of nano copper were also prominent in crustaceans [38]. It can also change movement parameters of fish spermatozoa as a study on *O. mykiss* shows [39] and cause critical changes in the reproductive organs and lower the survival rate of larvae and cause developmental damage [40,41,42]. Changes in gene expression induced by nano copper were tested among the other parameters as well.

In the study on *Epinephelus coioides* subjected to a treatment with copper nanoparticles and copper sulphate, it was shown that after the treatment with either form of CuNPs some genes involved in lipid transport and metabolism were up-regulated, such as those encoding apolipoprotein Eb, carnitine O-palmitoyltransferase 1 and acetyl-CoA acetyltransferase, complement component 3, complement component 8 subunit 2β, and component 9 gene. Some of the down-regulated genes were the cytochrome P450 gene, and sorbitol dehydrogenase or α/β hydrolase domain-containing protein 14B [43]. In the hepatic cells culture of *E. coioides* after treatment with CuNPs, the anti-oxidative-related genes (SOD Cu/Zn, SOD Mn, CAT (catalase), GPx4 (glutathione peroxidase)) were down-regulated, and the up-regulated genes included *p53*, *p38β*, TNF-α [44]. On the other hand, in *Oreochromis niloticus,* 50 mg L^−1^ caused elevation of the transcripts for SOD, CAT, and GP, but TNF- α also showed up-regulation. Additional up-regulated genes consisted of IL-1β, IL-12, Hsp70, and caspase3 [45].

When aquatic macrophyte *Elodea nuttalii* was subjected to a treatment with 1.4–2 mg L^−1^ CuNPs, it was discovered that the expression of the copper transporter COPT1 was down-regulated, which led to limiting the uptake of excess Cu [46].

Sublethal levels of nanocopper (1.5 mg L^−1^) in adult zebrafish induced up-regulation of HIF-1 (response to hypoxia), HSP70 (general stress response), and CTR (calcitonin receptor) [11].

Doses of 20 and 100 µg L^−1^ of CuNPs administered for seven days to *Cyprinus carpio* induced up-regulation of diphosphomevalonate decarboxylase and selenide/water dikinase-1, as well as down-regulation of ferritin heavy chain, rho guanine nucleotide exchange factor 17-like, and cytoglobin-1, respectively involved in the synthesis of isopentyl dipshosphate, transport of phosphorus-containing groups, ferrooxydation, and stimulation of formation of Rho-GTP [47].

Investigation showed that O*. mykiss* subjected to a treatment with CuNPs showed increased hepatic and intestinal levels of metallothionein (metal ions homeostasis), with much higher up-regulation in liver [48]. 

After 15 days of exposure of 10 µg L^−1^ of CuNPs to mussels *M. galloprovincialis*, 103 genes in the digestive glands and 119 genes in the gills showed different expression levels with a tendency of up-regulation of the expression in the gills and down-regulation of the expression in the digestive glans. Down-regulated genes included zinc-finger BED domain-containing protein 1 and caspase 1/7-1, whereas some of the up-regulated genes were S-glutathione transferase (defense against ROS), ATP synthase F0 subunit 6 (ATP synthesis), cathepsin L (protease), heat shock cognate 71, and precollagen-D [49].

### 2.3. Effects of other Nanoparticulate Metals on Gene Expression

Many other nanoparticulate metals are currently used in industry. Zinc nanoparticles (ZnNPs) are used in sunscreens because of their ability to absorb UV light, and they also show antibacterial properties. They are also extensively tested for their application in cancer therapy [50]. Apart from their use in healthcare, they can also be found in products such as rubber, paint, coating, and cosmetics [51]. 

Titanium nanoparticles (TiNPs) are suitable for use in pharmaceuticals, coatings, inks, and food products due to their preservative properties. They are also used in clothing and anti-fogging coating [52].

Gold nanoparticles (AuNPs) found use in the medical industry in the targeted delivery of drugs, optical bioimaging, diagnostics [53]. Outside the medical field, AuNPs are used as catalysts [54]. Another type of nanoparticles that are suitable for use in medical field are selenium nanoparticles (SeNPs) used mostly as drug carriers and cerium nanoparticles (CeNPs) [55,56].

Iron nanoparticles (FeNPs) can be used in water remediation, in dye removal, as antibacterial agents, and in medical field [57].

Though very versatile and very useful, these nanoparticles also pose a threat to aquatic organisms. 

A rare case of nanoparticle metal having a positive effect on an organism was discovered when *M. galloprovincialis* was treated simultaneously with cadmium and TiNPs. Cadmium alone induced up-regulation of *abcb1* gene (ATP Binding Cassette Subfamily B Member 1); however, when TiNPs were involved, the transcript level was lower [58]. In *Danio rerio* embryos, however, TiNPs negatively altered the expression of genes for such proteins as type I cytokeratin (regulation of kinases activity), cytochrome P450 and family 51 (involved in cholesterol synthesis), zona pellucida glycoprotein 3a.2 (component of zona pellucida), serum/glucocorticoid regulated kinase 1 (involved in cellular stress response), and prostaglandin D2 synthase (involved in smooth muscle contractions) and at the same time up-regulated the expression of, e.g., carboxyl ester lipase (which catalyzes fat and absorbs vitamins), as well as activin receptor 1ib (kinase) [59]. TiNPs may also cause up-regulation of metallothionein genes in *Danio rerio* embryos [25]. In the green algae *C. reinhardtii,* TiNPs caused up-regulation of 96 genes, and 80 genes were down-regulated [34].

In *Daphnia magna* exposed to sublethal concentrations of ZnNPs nanoparticles, genes for multicystatin (protease inhibition) were up-regulated in the concentration of both 2.2 mg L ^−1^ of ZnNPs and 9 mg L^−1^, whereas in the same concentrations, the ferritin (iron binding), C1q domain protein (activation of complement system), and nucleoside transporter genes became down-regulated [60].

Exposure of ZnNPs to zebrafish embryos caused up-regulation of catalase and Cu/Zn SOD after 48 hpf, but after 96 hpf, these genes were down-regulated; other up-regulated genes included metallothionein 2 after both 48 and 96 hpf, and *c-jun* after 96 hpf, whereas MxA and TNF showed down-regulation of expression. The same zinc nanoparticles influenced zebrafish eleuthero embryos differently when for catalase and Cu/Zn SOD genes, which became less expressed after TNF as well, while other genes showed similar expression as in other embryos [61]. Zhao et al. [62] found elevated transcript levels of pro-apoptotic *bax*, *puma,* and *apaf1* genes in the zebrafish embryos treated with ZnNPs, and at the same time, the anti-apoptotic *bcl-2* gene was under-expressed. 

In another study on developing zebrafish embryos treated with ZnNPs, it was found that 358 genes were up-regulated and 112 down-regulated. In the up-regulated group, there were such genes as *ogfrl2* (opioid growth factor receptor-like 2, involved in inhibition of DNA synthesis) and *cyb5d1* (cytochrome b5 domain-containing 1, involved in detoxification), and reduction of intelectin 2 (development and innate immunity) was reported [63].

In the testis of *C. carpio,* 100 µg L^−1^ ZnNPs suppressed the expression of steroidogenic enzyme genes (*20**β**-hsd*, *cyp19a1*) and transcription factors (*dmrt1*, activin β, *dax1*, *foxt2*, *ad4bp*, *wnt5*) [64].

Green algae, *C. reinhardtii,* when subjected to a treatment with ZnNPs, were characterized by 156 up-regulated and 29 down-regulated genes [34]. 

Dawood et al. [65,66] performed a study using SeNPs. Dietary SeNPs fed to *O. niloticus* increased expression of *TNF, IL-1β* genes, and *SOD*. O. niloticus fed with SeNPs had down-regulated *Hsp70* gene, a biomarker of shock by various stimulants to the organism [65,66,67].

In the developing eyes of the zebrafish embryos treated with sublethal doses of AuNPs functionalized with a cationic ligand N,N,N-trimethylammoniummethanethiol, significant repression of transcription factors pax6a, pax6b, otx2, and rx1 occurs, as well as of pigmentation transcription factor sox10 [68]. Spiked sediment containing gold nanoparticles induced up-regulation of *sod1, sod2, cox1* (mitochondrial respiration)*, mt2, gaad* (DNA repair), and *ache* (neurotransmission) genes in adult *Danio rerio* brain, but in the gills as well as in muscles, *sod2, cox1, gaad* and additionally *rad51* (DNA repair) genes showed decreased expression, and, also in the muscles, down-regulation of hsp70 was noted [69].

Teles et al. [70] performed a study in which sea bream (*Sparus aurata*) was exposed to 0.5 and 50 µg L^−1^ AuNPs. In the 0.5 µg ^−1^ AuNPs group elevation of the expression of genes involved in response to xenobiotics, such as selenium binding protein 1, was elevated. Other groups of genes that were up-regulated consisted of genes encoding proteins that show oxidoreductase activity, such as L2HGDH, SULT1A3, and CYP2N, which involved in immune response, including NLRP3, ST6GAL-1, and integrin β 1 binding protein 3, as well as DNA repair/apoptosis regulation genes, such as perforin 1 or MSH6. Higher transcripts of transcription, DNA processing and translational regulation genes were also found, i.e., MEF2A, EIF4E, and RPL13a. Other genes included, among others, those encoding ETNK1 (lipid metabolism), mitotin (cell adhesion), and talin 1 (cytoskeleton organization).

In the 0.5 µg L^−1^ AuNPs, down-regulation of certain genes was also discovered. Those included, among others, glucose dehydrogenase (carbohydrate metabolism), VPS4B (cell cycle regulation), c-ski (repressor of TGFb), and ccdc59 (transcription factor).

The second group, fish exposed to 50 µL L^−1^ AuNPs, were characterized by up-regulation of genes involved in response to xenobiotics (ABCF2), oxidoreductase activity, such as electron transport chains (UQCRFS1, UQCRH), and immune response (CCDC86). Genes involved in stress response also showed up-regulation, such as CHMP4C or HSC70, as well as genes involved in lipid/protein metabolism, i.e., farnesyl pyrophosphate synthetase, PITRM1, TMEM147, and PSMC3, as well as genes involved in transcription, DNA processing and translational regulation, such as exosome complex exonuclease RRP45, TBL3, WBP11.

Down-regulated genes in the second group of fish involved chemokine CK-1 (immune response), NGFR, programmed cell death 7, MAP2K7 (all involved in apoptosis), elastase-like serine protease, ACTN1 (cytoskeleton protein), bloodthirsty (erythrocyte differentiation), and ankyrin repeat-containing protein (transcription factor). 

SiNPs were also tested for changes in gene expression on *S. aurata*. A 100 mg L^−1^ measure of SiNPs triggered up-regulation of stress-induced *Hsp70* gene and cytokines genes, including TNF- α, IL-1 β, IL-12, and IL-8 in the gills and liver. In both organs, SOD gene responding to elevated ROS levels was found to be up-regulated, as well as CASP3 (caspase), which plays a role in apoptosis [71].

FeNPs nanoparticles (Fe_3_ONPs, magnetite) were found to change the expression of 5571 genes, which included 59% down-regulated genes and 41% up-regulated genes in the gills of adult zebrafish. More differently expressed genes—5571—were observed in the liver. In both liver and gills, transcript levels of genes involved in iron metabolism were elevated, including such proteins as ferroportin-1, alas2, TfR1b, and TfR2. The liver was also characterized by elevated levels of pro-apoptotic genes transcripts (*mt-nd4, mt-nd5, mt-cyb, cox17, cox6a1, mtco2,* and *mt-co3)* and up-regulation of *tsc22d3* (leucine zipper transcription factor), a marker of inflammation. The expression of genes related to DNA repair was suppressed in the liver [72].

In the gills, magnetite nanoparticles induced up-regulation of such genes as *abcb4* (stress response), *sgk1* (protein kinase), and *tsc22d3* (anti-inflammatory response), whereas *fen1* (DNA repair) and *sirt7* (epigenetic regulation) were down-regulated [72].

Another form of iron nanoparticles, γ-Fe_2_O_3_ (maghemite), administered to adult zebrafish, caused 953 genes in the liver to be expressed differently. Maghemite elevated the transcripts of antioxidant genes (*aldh5a1, gpx1a, gstm3, hsd3b7, hspb1, ugtc5c*) in the liver. Genes related to ion binding, translation, and ribosome biogenesis were up-regulated as well [73].

RNA-seq of *D. rerio* cells subjected to a treatment with carboxymethyl cellulose stabilized iron sulfide nanoparticles revealed 3200 down-regulated genes (55.2% of total differentially expressed genes), including *cmc4* (involved in mitochondrial protein import) and *pimr214* (Ser/Thr kinase activity), and 2593 up-regulated genes (44.8% of total differentially expressed genes), including such genes as *hist1h4l* (histone H4 gene), *hist1h4a* (histone H4 gene), *ighv-5* (V region of the variable domain of immunoglobulin heavy chains), and *dut* (deoxyuridine triphosphatase). Differentially up-regulated KEGG pathways included complement and coagulation cascades, serine, and threonine metabolism; PPAR signaling pathway; tryptophan metabolism; and fat digestion and absorption. Down-regulated KEGG pathways included cell cycle, DNA replication, pyrimidine metabolism, RNA transport and degradation, purine metabolism, and ubiquitin-mediated proteolysis [74]. 

In the study performed by Morel et al. [75], green algae, *C. reinhartdii,* were exposed to uncoated and coated either with polyacrylic acid (PAA) or citrate (Ci) CeNPs. The uncoated CeNPs group produced 688 DEGs, PAA-coated 315, and Ci-coated only 23. All types of CeNPs induced up-regulation of the genes involved in the xenobiotic resistance system. Uncoated and PAA-coated CeNPs caused over-expression of *FAP16* gene, whose protein product induces detachment of the flagella and at the same time cause under-expression of the *POC7* gene, which is related to flagella assembly. 

## 3. Conclusions

The growing interest in nanoparticulate metals in general industry inevitably leads to increasing environmental pollution by nanoparticles and ions released by them, which includes water pollution. This has led to a need to investigate the toxic effects of metal nanoparticles on aquatic organisms, and it was established that they have a gene-expression-modulatory effect.

AgNPs were discovered to trigger overexpression of genes involved in inflammation and apoptosis and under-expression of genes whose protein products are involved in body development. Genes involved in calcium-binding might show overexpression, whereas those that participate in DNA-dependent transcriptional regulation, cell proliferation, and differentiation might be expressed, which might lead to delayed development, and genes participating in oocyte differentiation might also be repressed. Expression of genes for oxyradical scavengers and enzymes with antioxidative properties in organisms subjected to treatment with AgNPs is not uniform; some organisms present over-expression of those genes, and some show repressed expression. mRNA levels of genes involved in protein catabolism, those encoding transcriptional factors and involved in ribosomal assembly might be under-expressed which might lead to unbalanced levels of different proteins (Table 1).

CuNPs were also discovered to have a modulatory effect on gene expression of aquatic organisms. As with AgNPs, anti-oxidative related genes and genes involved in general stress response might be either over- or under-expressed, and metallothionein mRNA levels are found to be elevated. CuNPs were also involved in elevating the mRNA levels of genes that participate in lipid transport and metabolism, as well as cell cycle regulation and induction of apoptosis (Table 2). 

Other nanoparticulate metals that were proven to have a modulatory effect on gene expression were ZnNPs, SiNPs, AuNPs, TiNPs, SeNPs, and CeNPs (Table 3).

ZnNPs had an up-regulating effect on genes involved in protease activity inhibition and response to oxidative stress. They also may elevate the transcripts levels of genes involved in binding of heavy metals and those that have pro-apoptotic activity. On the other hand, ZnNPs may suppress transcription of genes that play a significant role in immune responses.

SiNPs may induce expression of immune-response genes as well as general-stress-response genes and oxidative-stress-response genes. Transcripts of genes with pro-apoptotic activity may also be elevated. 

FeNPs trigger up-regulation of such groups of genes as genes involved in iron metabolism, stress response genes, anti-inflammatory genes, pro-apoptotic genes, and antioxidants and ROS scavengers. DNA repair genes may, however, be under-expressed. 

AuNPs may induce over-expression of genes involved in oxidoreductive activity, immune response, apoptosis regulation, lipid metabolism, cell adhesion, cytoskeleton organization, response to xenobiotics, and electron transport chain, whereas cell cycle regulation genes and transcription factor may be under-expressed. There is also inconsistency in some cases in changes of expression of some genes induced by AuNPs in different organs. In the brain, genes involved in response to ROS and DNA repair may be up-regulated, but in the gills, they might be down-regulated.

TiNPs disrupt the expression of genes that are involved in the regulation of kinase activity, cholesterol synthesis, and zona pellucida formation.

SeNPs may protect cells from the toxic effects of other metals but can also induce cytokines’ gene expression, as well as expression of various genes involved in general stress response.

CeNPs may induce detachment of flagella and inhibit flagellation, which is a marker of stress in flagellated organisms. 

Understanding the impact of nanoparticles on aquatic organisms and gene expression will help to assess the risk following their release into the water and could help minimize it. It is important to know exactly how extensive the toxic effects of metal nanoparticles on aquatic organisms are, so placing importance on the prevention of water pollution is justified. Knowing which genes specifically can be targeted by nanoparticles might be important as biomarkers in the evaluation of the pollution levels by molecular studies. It is limited, however, just to assess the changes in the expression of the genes and not of the synthesis of the proteins, which may be controlled on translational levels.

## Figures and Tables

**Table 1 ijerph-18-08361-t001:** Effects of silver nanoparticles on the genes expression of water organisms

Species Tested	Cell Tissue Target	Genes Targeted/Expression Changes	Gene Main Function	References
***Danio rerio***	Embryos	- *zgc:175127* ↑ - *sort1b* ↑ - *idh3g* ↑ - *thrab* ↑ - *lgals3l* ↑ - *caspb* ↑ - *zgc:114104* ↑ - *pvalb5* ↑ - *lnx1* ↑ - *per1b* ↓ - *nt5c3* ↓ - *mif4gdb* ↓ - *cbx7a* ↓ - *mep1a.2* ↓ - *slc45a3* ↓ - *myf5* ↓ - *irg1l* ↓	- Cellular component - Involved in lipid metabolism - carbohydrate metabolism - developmental processes - lectin - proteolysis - metal ion binding - calcium/calmodulin binding - notch signalling gene - response to light stimulus - pyrimidine nucleoside metabolism - regulation of translation - chromatin modification - proteolysis - transmembrane transport - muscle development - propionate metabolism	[22]
		- *pepd* ↑ - *sema3fb* ↑ - *map2k2a* ↑ - *anxa1b* ↑ - *mylpfb* ↑ - *fgf11a* ↑ - *cdh15* ↑	- peptidase - semaphoring - mitogen-activated kinase - annexin - myosin light chain - fibroblast growth factor - cadherin	
		- HIF4 ↑ - *Pxmp2* ↑	- response to hypoxia - channel forming protein in peroxisomal membrane	[16]
		- TGFβ ↑	- multifunctional cytokine	
		- *pitx2* ↑ - *acvr2b* ↑ - *smad1* ↑ - *inhβB* ↓	- establishment of left – right axis - activin receptor precursor - transcriptional modulator - subunit of both activin and inhibin	[19]
	Liver	- MTF-1 ↓ - TLR4 ↓ - IL1B ↓ - CEPB ↓ - TRF ↓ - TLR22 ↓ - cytochrome 1a ↓	- transcription factor - pattern recognition receptor - cytokine - transcription factor - transcription factor - patter recognition receptor - proapoptotic activity	[21]
		- proteasome subunit β type-1 ↓ - ribosomal protein S6 modification-like protein ↑	- subunit involved in protein degradation - ribosomal protein ↑	[18]
		- glutathione peroxidase 1a ↓ - SOD1 ↓ - Bax1 ↑ - Noxa ↑ - *p21* ↑ - catalase ↑	- response to ROS - response to ROS - proapoptotic activity - transcription factor - cyclin-dependent kinase inhibitor - response to ROS	[24]
	Overall impact on body	- Mt2 ↑	- binding of heavy metals ↑	[25]
***Oncorhynchus mykiss***	Overall impact on body	- *cyp1a2* ↑	- xenobiotics metabolism	[23]
	Gills	- MT ↑ - GST ↑ - SOD ↑	- binding of heavy metals - response to ROS - response to ROS	[27]
***Chironomus riparius***	Fourth instar larvae	- ribosomal protein gene L15 ↓ - gonadotrophin releasing hormone gene ↑ - Balbiani repeat gene 2.2 ↑ - GST	- ribosomal protein - stimulation of gonadotrophin release - secretory protein - response to ROS	[28,29]
	Egss Larvae pupae	- Hsp70 ↑ - CrRPL15 ↓	- general shock response - ribosomal protein	[30]
***Oryzias latipes***	Liver	- GST ↑ - Hsp70 ↑ - choriogenin↑ - vitellogenin ↑	- response to ROS - response to stressors - precursor of the inner layer subunit of the egg envelope - precursor of egg yolk	[31]
	Overall effect on body	- *cyp1a* ↓ - *NKA* ↓	- response to stressors - nucleotide binding	[17]
***Mytillus galloprovincialis***	Overall impact on body	- CYP4YI ↑ - cathepsin ↑ - GST ↓ - caspase ↓	- xenobiotic metabolism - protease - response to ROS - proapoptotic activity	[32]
***Strongylocentrotus droebachiensis***	Immune cells	Hsp70 Hsp60 ↑	- response to stressors	[20]
***Chlamydomonas reinhardtii***	Overall impact on organism	*- CYC6* *↑* *- FDX5* *↑* *- CTR2* *↑* *- CRD1* *↑*	- copper level regulation - reductive metabolism - copper transporter - copper assimilation	[33]

↑ - up-regulation of the gene; ↓ - down-regulation of the gene.

**Table 2 ijerph-18-08361-t002:** Effects of copper nanoparticles on the expression of genes of water organisms.

Species Tested	Cell and Tissue Target	Genes Targeted/Expression Changes	Gene Main Function	References
***Epinephelus coioides***	Overall impact on body	- apolipoprotein Eb ↑ - carnitine O-palmitoyltransferase 1 ↑ - acetyl-CoA acetyltransferase ↑ - complement component 3 ↑ - complement component 8 subunit β ↑ - complement component 9 ↓ - cytochrome p450 ↓ - sorbitol dehydrogenase ↓ - α/β hydrolase domain-containing protein 14 β ↓	-apolipoprotein precursor - formation of acyl carnitines - transfer of aminoacyl groups - element of immune response - element of immune response - element of immune response - xenobiotics metabolism - conversion of sorbitol into fructose - hydrolase	[45]
	Liver cells culture	- SOD Cu/Zn ↓ - SOD Mn ↓ - CAT ↑ - GPx4 ↑ - *p53* ↑ - *p38β* ↑ - TNFα ↑	- response to ROS - response to ROS - response to ROS - response to ROS - cell cycle regulation - mitogen-activated protein kinase - cytokine	[44]
***Danio rerio***	Overall impact on body	- HIF-1 ↑ - Hsp70 ↑ - CTR ↑	- response to hypoxia - response to various stressors - calcitonin receptor	[11]
***Cyprinus carpio***	Overall impact on body	- diphosphomevelonate decarboxylase ↑	- conversion of mevalonate 5-diphosphate into isopentyl diphosphate	[47]
		- selenide/water dikinase-1 ↑	- transfer of phosphorus-containing groups	
		- ferritin heavy chain ↓	- ferroxidase	
		- rho guanine nucleotide exchange factor 17-like ↓ - cytoglobin-1 ↓	- stimulation of the formation of Rho-GTP - response to ROS	
***Oncorhynchus mykiss***	Liver and gut	- metallothionein ↑	- heavy metals binding	[48]
***Mytilus galloprovincialis***	Digestive gland and gills	- S-glutathione transferase ↑ - cathepsin L ↑ - ATP synthase F0 synthase subunit ↑ - heat shock cognate 71 ↑ - precollagen – D ↑	- response to ROS - proteinase - subunit of ATP synthase - response to stress - protocollagen	[49]
***Elodea nuttalii***	Overall impact on the organism	- COPT1 ↓	- copper transporter	[46]

↑ - up-regulation of the gene; ↓ - down-regulation of the gene.

**Table 3 ijerph-18-08361-t003:** Effects of other metal nanoparticles on the expression of genes of water organisms

SPECIES TESTED	Nanoparticle Type	Cell and Tissue Target	Genes Targeted/Expression Changes	Gene Main Function	References
***Mytilus galloprovincialis***	TiNPs	Overall impact on body	- *abcb1* ↓	- ATP-binding cassette	[58]
***Danio rerio***	TiNPs	Embryos	- type I cytokeratin ↓ - cytochrome P450 and family 51 ↓ - zona pellucida glycoprotein 3a.2 ↓ - serum/glucocorticoid regulate kinase 1 ↓ - prostaglandin D2 synthase ↓ - carboxyl ester lipase ↑ - activin receptor lib ↑	- regulation of kinase activity - cholesterol synthesis - zona pellucida component - cellular stress response - smooth muscle contraction - fat and vitamin absorption - kinase	[59]
***Daphnia magna***	ZnNPs	Overall impact on body	- multicystatin ↑ - ferritin ↓ - C1q domain protein ↓ - nucleoside transporter ↓	- protease inhibition - iron binding - activation of complement system - nucleoside transport	[60]
***Danio rerio***	ZnNPs	Embryos	- Cu/Zn SOD ↑ - metallothionein ↑ - *c-jun* ↓ - MxA ↓ - TNF-α ↓	- response to ROS - binding of heavy metals - transcription factor domain - GTP-metabolizing protein - cytokine	[61]
			- *bax* ↑ - *puma* ↑ - *apaf1* ↑ - *bcl-2* ↓	- proapoptotic activity - proapoptotic activity - proapoptotic activity - anti-apoptotic activity	[62]
			- *ogfrl2* ↑ - *cyb5d1* ↑ - intelectin 2 ↓	-inhibition of DNA synthesis - detoxification - development/innate immunity	[63]
***Danio rerio***	ZnNPs	Eleuthero-embryos	- CuZn SOD ↓ - TNF-α ↓	- response to ROS - cytokine	[61]
***Oryzias niloticus***	SeNPs	Overall impact on body	- TNFα ↑ - IL-1β ↑ - SOD ↑ - Hsp70 ↓	- cytokine - cytokine - response to ROS - response to various stressors	[65,66,67]
***Danio rerio***	AuNPs	Brain	- *sod1* ↑ - *sod2* ↑ - *cox1* ↑ - *mt2* ↑ - *gaad* ↑ - *ache* ↑	- response to ROS - response to ROS - mitochondrial respiration - response to heavy metals - DNA repair - neurotransmission	[69]
		Gills	- *sod2* ↓ - *cox1* ↓ - *raad51* ↓ - *gaad* ↓	- response to ROS - mitochondrial respiration - DNA repair - DNA repair	
***Sparus aurata***	AuNPs	Overall impact on body	- selenium binding protein 1 ↑ - L2HGDH ↑ - SULT1A3 ↑ - CYP2N ↑ - NLRP3 ↑ - ST6GAL ↑ - integrin β 1 binding protein 3 ↑	- response to xenobiotics - oxidoreductive activity - oxidoreductive activity - oxidoreductive activity - immune response - immune response - immune response	[70]
			- perforin 1 ↑ - MSH6 ↑ - MEF2A ↑ - EIF4E ↑ - RPL13a ↑ - ETNK1 ↑ - mitotin ↑ - talin1 ↑ - glucose dehydrogenase ↓ - VPS4B ↓ - *c-ski* ↓ - *ccdc59* ↓ - ABCF2 ↑ - UQCRFS1 ↑ - UQCRH ↑ - CCDC86 ↑ - CHMP4C ↑ - H5C70 ↑ - Farnesyl pyrophosphate synthethase ↑	- DNA repair/apoptosis - DNA repair/apoptosis - apoptosis regulation - apoptosis regulation - apoptosis regulation - lipid metabolism - cell adhesion - cytoskeleton organization - carbohydrate metabolism - cell cycle regulation - TGFβ repressor - transcription factor - response to xenobiotics - electron transport chain - electron transport chain - immune response - stress response - stress response - lipid/protein metabolism	
			- PITRM1 ↑ - TMEM147 ↑ - PSMC3 ↑ - exosome complex exonuclease RRP45 ↑	- lipid/protein metabolism - lipid/protein metabolism - lipid/protein metabolism - gene expression control	
			- TBL3 ↑ - WBP11↑	- gene expression control - gene expression control	
			- chemokine CK-1 ↓ - NGFR ↓	- immune response - apoptosis induction	
			- programmed cell death 7 ↓ - MAP2K7 ↓ - elastase-like serine protease ↓ - ACTN1 ↓ - bloodthirsty ↓ - ankyrin repeat-containing protein ↓	- apoptosis induction - apoptosis induction - protease - cytoskeleton protein - erythrocyte differentiation - transcription factor	
***Danio rerio***	FeNPs	Liver	- *aldh5a1* ↑ - *gpx1a* ↑ - *gstm3* ↑ - *hsd3b7* ↑ - *hspbl* ↑ - *ugt5c* ↑	- response to ROS - response to ROS - response to ROS - response to ROS - response to ROS - response to ROS	[73]
		Livergills	- *mt-nd4* ↑ - *mt-nd5* ↑ - *mt-cyb* ↑ - *cox17* ↑ - *cox6a1* ↑ - *mtco2* ↑ - *mt-co3* ↑ - ferroportin-1 ↑ - *alas2* ↑ - TfR1 ↑ - TfR2 ↑ - *cyp1a* ↑ - *abcb4* ↑ - *fen1* ↓ - *sgk1* ↓ - *tsc22d3* ↑ - *sirt7* ↓ - *stat2* ↓	- pro-apoptotic - pro-apoptotic - pro-apoptotic - pro-apoptotic - pro-apoptotic - proapoptotic - proapoptotic - iron metabolism - iron metabolism - iron metabolism - iron metabolism - response to stressors - response to stressors - DNA repair - protein kinase - anti-inflammatory response - epigenetic regulation - signal transductiontranscription activation	[72]
		Overal impact on body	- *cmc4* ↑ - *pimr214* ↑ - *hist1h4l* ↓ - *hist1h4a* ↓ - *ighv-5* ↓ - *dut* ↓	- mitochondrial protein import - Ser/Thr kinase - histone H4 gene - histone H4 gene - part of immunoglobulin - deoxyuridine triphosphatase	[74]
***Oreochromis niloticus***	SiNPs	Gills	- Hsp70 ↑ - TNF- α ↑ - IL-1 β ↑ - IL-8 ↑ - CASP3 ↑	- response to various stressors - cytokine - cytokine - cytokine - pro-apoptotic activity	[71]
		Liver	- SOD ↑ - Hsp70 ↑ - IL-1 β ↑ - IL-8 ↑ - TNF- α ↑ - CASP3 ↑	- response to ROS - response to various stressors - cytokine - cytokine - cytokine - pro-apoptotic activity	
***Chlamydomonas reinhardtii***	CeNPs	Overall impact on organism	- FAP16↑ - POC7↓	- flagella detachment - flagella assembly	[75]
***Cyprinus carpio***	ZnNPs	Testis	- *20 β-hsd* *- cyp19a1* - *dmrt1*	- steroidogenic enzyme - steroidogenic enzyme - transcription factor	[64]
			- activin β - *dax1* - *foxt2* - *ad4bp* - *wnt5*	- transcription factor - transcription factor - transcription factor - transcription factor - transcription factor	

↑ - up-regulation of the gene; ↓ - down-regulation of the gene.

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
