# Peer review of "Changes of Gene Expression Patterns from Aquatic Organisms Exposed to Metal Nanoparticles"

_ijerph, 2021, doi:10.3390/ijerph18168361_

Round 1

Reviewer 1 Report

This study made a review of changes of gene expression patterns from aquatic organisms exposed to metal nanoparticles. I should appreciate the authors' time and patient to come up with some results. However, there are several problems that deduct from the quality of this manuscript. Below are several comments on this work. 1. I cannot find any novel insights from this review. However, the paper does not go beyond established knowledge in this area in its critical evaluation. 2. Could you find any bulk or single-cell RNA-seq datasets to analyze the differential expression gene sets? 3. In Conclusion, there was no mention of the limitations of the study. 4. The authors should proofread the English writing to improve the study.

Author Response

Thank you for your valuable review. Changes have been made to the MS following your comments.

  1. I cannot disagree with this statement. The review was written to summarize the available knowledge in this field focusing more on nanoparticles and genes rather than taxonomic groups. It may be a valuable tool to show what has been done and what needs to be done in this area.
  2. Papers in which RNA-seq was performed were added.
  3.  The conclusion has been expanded
  4. The MS has been proofread by a native speaker from Ireland.

Reviewer 2 Report

This review aims to summarize the effects of metal nanoparticles on changes in gene expression of aquatic organisms. The topic is very interesting and it is important to understand the consequences of using metal nanoparticles and summarize current knowledge from this area of research. However, I have major concerns about the huge amount of mistakes regarding citations. E.g. many citations are missing in the list of References, some mentioned references are used incorrectly. It was many times not possible to verify the information because of the wrong citation. All these mistakes must be very precisely corrected. I would also recommend using some reference manager software to avoid the appearance of many mistakes. A large number of citation errors decrease the credibility of the paper.

The highlights should be formulated exactly to be clear e.g. which organism concerns the changes in the gene expression, which transcription factors were affected.

Here, I mention some examples of citation mistakes, but there are many more of them that need to be corrected.

Line 33: I did not find reference De et al. 2008 on the list of references.

Line 39:  « Khan et al 2017 » should be « Khan et al 2019 ».

Line 44: « (Nel 2005) (Huang et al. 2010) » should be « (Nel 2005, Huang et al 2010) ».

Moreover, Huang et al 2010 is not on the list of references.

Lines 46-47: Huang et al 2010 and Knaapen et al 2004 are not on the list of references.

Lines 51, 52: Knaapen et al 2004 and Smith et al 2001 are not on the list of references.

Line 61: I did not find reference Van Hoecke et al 2009 on the list of references. « Griffit et al 2007 » should be « Griffitt et al 2007 ».

Line 73: Reference Kwok et al 2016 is missing.

Line 105: Reference Cheng et al 2020 is not on the list of references.

It is not clear that whole part (lines 77-98) is summarizing the results of Park and Yeo.

Line 136: I did not find Gopalakrishnan on the list of references. 

Lines 302-306: There is a description of the effects of SiO nanoparticles on gene expression, but the reference (Abdel-Latif et al 2021) relates to copper oxide nanoparticles.

Lines 307-315: There is no reference mentioned.

All first-time used abbreviations should be explained (E.g. Line 38: NPs, Line 245: hpf).

Line 30: « use f such as » should be « use such as ».

Line 235: ex. ?

Line 236:  « lib » should be « Iib ».

Author Response

Thank you for your valuable review. The following changes have been made:

  1. Citations have been corrected using proper software and journal style template. It has been checked that the citations correspond with the text.
  2. The abbreviations have been explained and mistakes corrected

Reviewer 3 Report

This review paper discussed the effects of metal nanoparticles on the expression of various genes in aquatic organisms from various taxonomic groups. Nanometals were revealed to have a modulatory effect on gene expression in a variety of aquatic organisms. The topic is within the scope of this journal and the presentation is in a good manner. Therefore, I recommended the publications of the paper after major revision according to given my comments.

  • The abstract is not clear. Please add the aim and objective of the review.
  • Please write the aim and objective of this review at end of the introduction.
  • If possible add some more metal nanoparticles on gene expression.
  • In Conclusion, the authors should add the significance of this research, and its potential practical application.
  • English of the MS needs to be greatly improved. The English of the whole article has to be checked carefully to eliminate linguistic errors.
  • Gene names should be in italics. Some places in MS are missing. Please check the entire MS.
  • Please read the authors' instructions and follow the MS preparations. The present form of presentation is not this journal format.

Author Response

Thank you for your valuable review. The changes have been made following your comments:

  1. Abstract and introduction have been expanded
  2. Cerium nanoparticles and another type of iron nanoparticles were added
  3. Conclusion has been updated to contain missing information
  4. MS has been proofread by a native English speaker
  5. Proper software was used to update the citations, so they meet the journal's expectations

Round 2

Reviewer 1 Report

The authors have addressed all my comments.

Author Response

The reviewer has not submitted any more questions.

Reviewer 2 Report

The authors have not addressed my corrections.

In general, the way the authors answer my criticism is not acceptable. E.g. I asked for improvement of the Highlights section and I agree that it is not necessary to be part of this Ms, but I received no answer and this part just disappeared.

The references do not seem to be corrected by software as the authors claim. Can you specify which software? I spent the time to select many mistakes in citations, but it was not corrected. E.g.: Line 38 – reference [3] Correct names of authors are missing even I was explicitly asking for that – this is also confirmation that references are not processed by reference manager software.

The source of information used in the tables is not clear for readers, because there are e.g. 6 mentioned articles, but individual genes usually studied 1-2 of them. Can you keep the style of using references in tables as it was in the previous version of Ms? There should be clear who detected which expression changes. Moreover, there are some mistakes that need to be corrected, e.g. in Table 1, reference no 25 is missing. The level of MT2 was studied in Danio rerio embryos instead of the overall impact on the body. In Oncorhynchus mykiss, increased expression of cyp1a2 was not studied by Johari et al. 2015. Please, check properly all Tables and use correct references.

I also asked for precise referring to used sources, but it was not improved. E.g. Lines 318-326: There is again no reference mentioned for this paragraph. No reference for paragraph lines 96-102.

Also, the abbreviations are not corrected properly, e.g. AgNT, AgNC.

Author Response

Thank you for your response. I apologise for my previous response, I did not mean any disrespect.

1. Apologies for failing to mention the changes in the highlights. They were removed as they were not required by the Journal and they seemed redundant. 2. The software used is MS Word with the Mendeley plug-in. Citation [3] was written like this because Mendeley uploaded the citations from the website this way. This was corrected in the MS.

3. The citations and publications in the tables now correspond with the genes from which they were cited. Citations 65-67 were left as they were because they mention the same genes in some places.

4. Citations in this paragraph were added.

5. The abbreviations were corrected.

Reviewer 3 Report

Requested corrections were completed.

Author Response

(The authors gave the same response as above.)
